# Monoclonal Antibodies Specific for SARS-CoV-2 Spike Protein Suitable for Multiple Applications for Current Variants of Concern

**DOI:** 10.3390/v15010139

**Published:** 2022-12-31

**Authors:** Mahali S. Morgan, Kexin Yan, Thuy T. Le, Ryan A. Johnston, Alberto A. Amarilla, David A. Muller, Christopher L. D. McMillan, Naphak Modhiran, Daniel Watterson, James R. Potter, Julian D.J. Sng, Mary Lor, Devina Paramitha, Ariel Isaacs, Alexander A. Khromykh, Roy A. Hall, Andreas Suhrbier, Daniel J. Rawle, Jody Hobson-Peters

**Affiliations:** 1School of Chemistry and Molecular Biosciences, University of Queensland, St Lucia, QLD 4072, Australia; 2Inflammation Biology, QIMR Berghofer Medical Research Institute, Herston, QLD 4006, Australia; 3Global Virus Network Centre of Excellence, Australian Infectious Diseases Research Centre, Brisbane, QLD 4072 and 4029, Australia

**Keywords:** monoclonal antibody, SARS-CoV-2, immunohistochemistry, immunofluorescence, Western blotting, lateral flow assays, immunotherapy

## Abstract

The global coronavirus disease 2019 (COVID-19) pandemic caused by the severe acute respiratory syndrome coronavirus 2 (SARS-CoV-2) has spawned an ongoing demand for new research reagents and interventions. Herein we describe a panel of monoclonal antibodies raised against SARS-CoV-2. One antibody showed excellent utility for immunohistochemistry, clearly staining infected cells in formalin-fixed and paraffin embedded lungs and brains of mice infected with the original and the omicron variants of SARS-CoV-2. We demonstrate the reactivity to multiple variants of concern using ELISAs and describe the use of the antibodies in indirect immunofluorescence assays, Western blots, and rapid antigen tests. Finally, we illustrate the ability of two antibodies to reduce significantly viral tissue titers in K18-hACE2 transgenic mice infected with the original and an omicron isolate of SARS-CoV-2.

## 1. Introduction

The global SARS-CoV-2 pandemic has spawned a considerable demand for new monoclonal antibodies (mAbs) for a range of diagnostic and research applications. These include immunohistochemistry (IHC) [1,2], immunofluorescence assays [3,4], lateral flow rapid antigen tests [5], ELISA [6,7], and Western blotting [8,9]. In addition, several mAbs have also been developed by a range of companies [10] for use in the treatment of COVID-19 patients [11] and for prophylactic prevention of COVID-19 [12,13], although many commercially developed mAbs are no longer effective against new strains of concern [14].

Herein we describe the generation of a panel of mAbs with applications in a range of assay systems. Of particular interest was SCV2-1E8, a mAb that showed excellent utility for IHC for both original and omicron isolates, as well as two other mAbs, SCV2-5A1 and SCV2-3H9 that showed therapeutic potential against original and omicron variants in the K18-hACE2 mouse model.

## 2. Materials and Methods

### 2.1. Ethics Statements

Mouse experiments were approved by the QIMR Berghofer MRI Biosafety Committee and Animal Ethics Committee (project P3600) and conducted in accordance with the “Australian Code for the care and use of animals for scientific purposes” as defined by the National Health and Medical Research Council of Australia. The conditions the mice were kept are as follows: light = 12:12 h dark:light cycle, 7:45 a.m. sunrise and 7:45 p.m. sunset, 15 min light-dark and dark-light ramping time. Enclosures: M.I.C.E cage (Animal Care Systems, Centennial, CO, USA). Ventilation: 100% fresh air, eight complete air exchange/h/rooms. In-house enrichment: paper cups (Impact-Australia, Sydney, Australia), tissue paper and/or cardboard rolls. Bedding: PuraChips (Able scientific) (aspen fine). Food: Double bagged Norco rat and mouse pellet (AIRR, Darra, Queensland, Australia). Water: deionized water acidified with HCl (pH = 3.2).

Collection of nasal swabs from COVID-19 patients and isolation of virus at QIMR Berghofer MRI was approved by the QIMR Berghofer MRI Human Research Ethics Committee (P3600). Patients were asked to sign an Information and Consent form, and took their own nasal samples using swabs from RAT tests. Virus isolates were deidentified and no patient data was recorded except that they had illness associated with positive PCR and/or RAT COVID-19 test(s).

### 2.2. Biosafety PC3 Certifications

All work with infectious SARS-CoV-2 virus was conducted in a biosafety level-3 (PC3) facility at the QIMR Berghofer MRI (Australian Department of Agriculture, Water and the Environment certification Q2326 and Office of the Gene Technology Regulator certification 3445), or at The University of Queensland (Australian Department of Agriculture, Water and the Environment certification Q2024 and Office of the Gene Technology Regulator certification 195).

### 2.3. SARS-CoV-2 Viruses and UV Inactivation

An original (ancestral) strain isolate, SARS-CoV-2_QLD02_, was isolated from patient’s nasopharyngeal aspirates by inoculation of Vero E6 cells, with virus stocks produced in Vero E6 cells (ATCC, CRL-1586). The isolate was kindly provided by Dr Alyssa Pike and Frederick Moore (Queensland Health, Brisbane, Australia). SARS-CoV-2_QLD02_ was sequenced by Dr D. Warrilow (Queensland Health, Brisbane, Australia); hCoV-19/Australia/QLD02/2020, GISAID Accession ID; EPI_ISL_407896. The omicron isolate, SARS-CoV-2_QIMR01_, was isolated as above at QIMR Berghofer MRI and sequenced (GenBank Accession number ON819429 and GISAID EPI_ISL_13414183). SARS-CoV-2_QIMR01_ (B.1.1.529), belongs to the BA.1.17 lineage [15]. Virus stocks were propagated in Vero E6 cells and were checked for mycoplasma (MycoAlert, Lonza, Basel, Switzerland) [16] and endotoxin [17].

Viral supernatants were UVC-inactivated in 6 well plates (3 mL of RPMI 1640 supplemented with 10% FBS per well) using an UVC Hoefer ultraviolet cross-linker (4–4.7 mW/cm^2^ for 5 min). The UVC dose was confirmed using a UVC Light Meter (Lutron UVC-254A) and inactivation confirmed by CCID_50_ assay [18].

### 2.4. Immunization and mAb Production and Purification

Two female C57BL/6J mice (≈5 months old) were immunized twice s.c. with 25 µg UVC-inactivated SARS-CoV-2_QLD02_, separated by 3 weeks. The inactivated virus was partially purified using a 20% sucrose cushion [19], protein concentration determined by Bradford Assay (Bio-Rad, Hercules, CA, USA), and was adjuvanted with QS-21 and 3-O-desacyl-4′-monophosphoryl lipid A as described [20]. After ≈2 months neutralization activity of serum against SARS-CoV-2_QLD02_ was confirmed [19,21]. Mice received a further 87 µg of UVC-inactivated, sucrose cushion purified, SARS-CoV-2_QLD02_ in PBS (no adjuvant), delivered i.v. and 4 days later mice were euthanized, and spleens harvested. Fusions were undertaken as described previously [22]. Briefly, spleens were harvested under sterile conditions and processed by passing through a 70 µm cell strainer with 10 mL RPMI 1640. Fusion was induced between spleen cells and NS0 myeloma cells (European Collection of Cell Cultures, Salisbury, UK) by the drop-wise addition of PEG-1500 (1 g/mL in RPMI 1640) at 37 °C. Fused cells were incubated overnight at 37 °C with 5% CO_2_ in a conditioned flask. Selection for hybridomas was performed by the addition of HAT supplement (hypoxanthine, aminopterin, thymidine; Hybri-Max, Sigma-Aldrich, St. Louis, MO, USA) and cells were plated in 96-well plates. Hybridomas secreting SARS-CoV-2-reactive mAbs were detected by ELISA using acetone-fixed SARS-CoV-2_QLD02_-infected Vero E6 cells. The infected cell monolayers were fixed using a solution of 80% acetone/20% PBS and placed at −20 °C for >1 h.

To verify the detection of a variety of SARS-CoV-2 proteins by ELISA using acetone-fixed SARS-CoV-2_QLD02_-infected Vero E6 cells, as outline above, an ELISA was performed with two nucleocapsid (N)-specific (C2, E2 [23]) and two spike-specific (MR17, Nb20 [24,25]) nanobodies that had been recombinantly expressed with a dimeric Fc tag as previously described [23]. The ELISA was performed as outlined in 2.10 below by titrating the nanobodies and detecting with horse radish peroxidase-conjugated goat anti-human IgG (Dako, P0214, Glostrup, Denmark).

Antibody isotyping was performed using Mouse Monoclonal Antibody Isotyping Reagents (Sigma-Aldrich, ISO2, St. Louis, MO, USA), as per the manufacturer’s instructions and using fixed SARS-CoV-2-infected Vero E6 cell monolayers as described above.

A subset of the monoclonal antibodies was purified by protein G immunochromatography (HiTrap Protein G HP, Cytiva, Marlborough, MA, USA) as per the manufacturer’s instructions and buffer-exchanged into PBS. For mouse protection studies, the mAbs were sterile-filtered through a 0.22 µm filter [22].

### 2.5. Infection of Mice with SARS-CoV-2 for IHC

Three mouse models were used to provide tissues for IHC (i) C57BL/6J mice infected with SARS-CoV-2_QIMR01_, (ii) SARS-CoV-2_QLD02_ infected K18-hACE2+/− mice that were originally purchased from The Jackson Laboratory (Bar Harbor, ME, USA) (B6.Cg-Tg(K18-ACE2)2Prlmn/J) and a colony maintained in-house by backcrossing onto C57BL/6J mice [26,27,28] and (iii) mACE2-hACE2 mice infected with SARS-CoV-2_QLD02_ [28]. Lightly anesthetized mice received intrapulmonary inoculations, 50 µL via the intranasal route as described [15,19]. The anesthesia was 3% isoflurane (Piramal Enterprises Ltd., Andhra Pradesh, India) delivered using The Stinger, Rodent Anesthesia System (Advanced Anaesthesia Specialists/Darvall, Gladesville, NSW, Australia). Mice were euthanized using CO_2_ on the indicated day post infection.

### 2.6. Immunohistochemistry

Lungs were fixed in >15 volumes of 10% formalin, and embedded in paraffin wax. For IHC, sections were affixed to positively charged adhesive slides and air-dried overnight at 37 °C. Sections were dewaxed and rehydrated through xylol and descending graded alcohols to Tris buffered saline (TBS) pH 7.6. Endogenous peroxidase activity was blocked by incubating the sections in 0.5% hydrogen peroxide in methanol for 10 min followed by three rinses in distilled water. Sections were transferred to Dako Epitope Retrieval Buffer and subjected to heat antigen retrieval (125 °C for 5 min) using the Biocare Medical de-cloaking chamber, and slides allowed to cool for 20 min before transferring to TBS. After washing in TBS 0.05% Tween 20, Biocare Medical Background Sniper + 2% BSA was applied for 15 min. After washing, AffiniPure Fab Fragment Donkey anti-mouse IgG (Jackson Immunoresearch, Philadelphia, PA, USA) diluted 1:50 in Biocare Medical Rodent block M was added for 60 min. After washing, Innovex Fc Receptor blocker was applied for 15 min. After washing, SCV2-1E8 hybridoma culture supernatant (≈20 µg/mL IgG) was diluted 1:70 in Biocare Medical Da Vinci Green diluent and applied overnight at room temperature. Sections were washed three times and goat anti-mouse IgG HRP (Perkin Elmer, Waltham, MA, USA) diluted 1:500 in 0.05% Tween 20 was applied at room temperature for 60 min. Sections were washed three times and color was developed in NovaRed (HRP) (Vector Laboratories, Newark, CA, USA) for 5 min. Sections were washed in gently running tap water for 5–10 min to remove excess chromogen. Sections were lightly counterstained in Mayers’ hematoxylin, and then dehydrated through ascending graded alcohols, cleared in xylene, and mounted using DePeX. All slides were scanned using Aperio AT Turbo (Aperio, Vista, CA, USA) and analyzed using Aperio ImageScope software (Leica Biosystems, Mt Waverley, Australia) (v10).

### 2.7. Indirect Immunofluorescent Assay Staining

UV-sterilized 13 mm coverslips were seeded with Vero E6 cells (10^5^ per 24 well) and incubated overnight in 1 mL RPMI 1640 supplemented with 5% fetal bovine serum (FBS) (Gibco, New York, NY, USA) previously tested for endotoxin contamination. Cells were infected with SARS-COV-2_QLD02_ (MOI ≈ 0.2) and cultured for 24 h. The medium was removed and cells were fixed in 100% ice-cold methanol for 2 min, washed in PBS and blocked with RPMI 1640 supplemented with 5% FBS for at least 1 h at room temperature. The blocking medium was removed and the primary monoclonal antibody (diluted in RPMI1640 supplemented with 0.5% FBS) was added for ≈1 h. After washing in PBS, the secondary antibody, goat anti-mouse IgG (H+L) Alexa Fluor plus 488 (Invitrogen, Waltham, MA, USA) (1 in 500 dilution in RPMI 1640 supplemented with 0.5% FBS) was added for ≈1 h. After washing in PBS, the coverslip was inverted and mounted on a glass slide using ProLong™ Glass Antifade Mountant with DAPI (Invitrogen). Staining was viewed using Zeiss LSM 780 NLO confocal microscope and the A488 nm laser.

### 2.8. MAb Neutralization Assay

Neutralizing antibody titers were determined as described [19,21]. Briefly, stock solutions of purified mAbs were serially diluted in RPMI1640 supplemented with 2% FBS in 96 well plates. For each mAb dilution, 50 µL/well was incubated with 50 µL of 100 CCID_50_ of SARS-CoV-2_QLD02_ in RPMI 1640 supplemented with 2% FBS, and incubated for 2 h at 37 °C. Vero E6 at 10^4^/well in RPMI 1640 supplemented with 10% FBS was then added in 100 µL. After 4 days of incubation, viral cytopathic effects (CPE) were determined by crystal violet staining and measurement of OD_590_ [18].

### 2.9. Production of Recombinant Spike Proteins

Recombinant spike proteins were generated as described [29]. The plasmid encoding SARS-CoV-2 S HexaPro was a gift from J. McLellan (Addgene plasmid no. 154754; https://www.addgene.org/154754/, accessed on 15 December 2022). This spike protein, encoding for the spike from the original strain, Wu-Hu-1 isolate (GenBank ID MN908947, was mutated to remove the furin cleavage site (amino acids RRAR to GSAG), and six proline substitutions that stabilized the proteins in a pre-fusion conformation (F817P, A892P, A899P, A942P, K986P, V987P) [30]. In addition to the original recombinant spike [29], trimeric HexaPro S protein from SARS-CoV-2 variants of concern, also encoding the furin cleavage site mutation and stabilizing proline substitutions, were developed; alpha (B.1.1.7; ∆H69, ∆V70, ∆Y144, N501Y, A570D, D614G, P681H, T716I, S982A, and D1118H), beta (B.1.351; L18F, D80A, D215G, ∆L242, ∆A243, ∆L244, R246I, K417N, E484K, N501Y, D614G, A701V), gamma (P.1; L18F, T20N, P26S, D138Y, R190S, K417T, E484K, N501Y, H655Y, T1027I), delta (B.1.617.2; T19R, G142D, ∆156E, ∆157F, R158G, L452R, T478K, D614G, P681R, and D950N) and omicron (BA.1; A67V, ∆H69, ∆V70, T95I, ∆G142, ∆V143, ∆Y144, Y145D, ∆N211, L212I insertion of EPE at 214, G339D, S371L, S373P, S375F, K417N, N440K, G446S, S477N, T478K, E484A, Q493R, G496S, Q498R, N501Y, Y505H, T547K, D614G, H655Y, N679K, P681H, N764K, D796Y, N856K, Q954H, N969K, L981F). DNA plasmids were used to transfect Expi293-F cells and spike proteins were purified using an immunoaffinity column containing the S-specific mAb 2M-10B11 or by Strep-Tactin XT resin (IBA) in the case of the omicron HexaPro spike [29,30].

Recombinant spike RBD and N-terminal domain (NTD) proteins were produced as previously described [29]. Briefly, plasmids containing SARS-CoV-2 (Wu-Hu-1) RBD and NTD fused to a human Fc were transfected into ExpiCHO-S cells (Thermo Fisher Scientific, Waltham, MA, USA) as per the manufacturer’s instructions. Culture supernatants were harvested and the protein purified via Protein A affinity chromatography.

### 2.10. ELISAs

Recombinant protein ELISAs were performed on Microlon high binding plates (Greiner, Kremsmünster, Austria), coated with 2 µg/mL recombinant spike, RBD and NTD proteins in PBS overnight at 4 °C. Plates were washed with PBS containing 0.05% Tween 20 (PBST) prior to the addition of blocking buffer (0.05 M Tris-HCl, pH 8.0, 1 mM EDTA, 0.15 M NaCl, 0.05% vol/vol, Tween 20, and 0.2% wt/vol casein) for 1 h at room temperature. Primary antibodies were applied as hybridoma culture supernatant, or purified preparations and incubated for 1 h at 37 °C. The mAbs were assessed in triplicate and applied as five-fold serial dilutions in blocking buffer starting at 10 µg/mL for purified mAbs, or undiluted hybridoma culture supernatant. For the RBD ELISA, purified mAbs were diluted to 100 µg/mL followed by 10-fold serial dilutions. Following three washes with PBST, horseradish peroxidase (HRP)-conjugated goat anti-mouse Ig (P0447, Dako; diluted 1/3000) was added for 1 h at 37 °C. Plates were washed six times with PBST prior to applying substrate solution (1 mM 2,2-azino-bis(3- ethylbenzothiazoline-6-sulfonic acid) (ABTS, Sigma-Aldrich) and 3 mM H_2_O_2_ in a buffer prepared by mixing 0.1 M citric acid with 0.2 M Na_2_HPO_4_ to give a pH of 4.2 and incubating for 1 h at room temperature in the dark. Absorbance was measured at 405 nm. For fixed cell ELISAs, mAbs were assessed as above using 100% acetone-fixed SARS-CoV-2_QLD02_-infected Vero E6 cells

Control mAbs included isotype controls C.10C1 (anti-Casuarina virus M protein, IgG2b [31]) and 4G2 (anti-dengue virus, IgG1 [32]). Recombinant expressed mouse anti-SARS-CoV-2 mAb CR3022 was used as a positive control and produced as previously described [19,26].

### 2.11. Lateral Flow Antigen Detection

Purified mAb SCV2-7E9 was conjugated to 40 nm carboxyl gold nanoparticles (NanoComposix, San Diego, CA, USA) as per manufacturer’s instructions. The conjugated gold was applied to pre-blocked conjugate pads (STD17, 370 µm, Cytiva), dried for 50 min at 37 °C and stored in desiccant until use. Pre-laminated FF120HP nitrocellulose membranes (Cytiva) were striped with purified mAb SCV2-6A11 at 1 mg/mL as the capture mAb, and anti-mouse IgG (Dako 1.5 mg/mL) as the control. The reagents were applied at 0.5 µL/cm using a Biodot AD3060 membrane striping machine.

Assembled lateral flow strips were assessed with 5-fold dilutions of original, delta and omicron recombinant spike proteins ranging from 1000 ng to 10 ng and chased with running buffer (10 mM PBS, 1% Tween-20). Strips were imaged with a colorimetric reader (Lumos Diagnostics, Carlsbad, CA, USA) after 10 min. Each sample was run in triplicate and capture line intensity graphed.

### 2.12. Western Blotting

Western blots were conducted as previously described [33] using lysates of mock-infected Vero E6 cells and Vero E6 cells infected with SARS-CoV-2_QLD02_. Briefly, lysates were run reduced (10 mM DTT, 90 °C for 3 min) or unreduced on NuPage 4–12% Bis-Tris protein gel (Invitrogen) at 175 V for 30 min, before being transferred to nitrocellulose membranes at 30 V for 60 min. Membranes were blocked in ELISA blocking buffer overnight at 4 °C. Membranes were probed with mAbs diluted in blocking buffer for 1 h before being washed with Tris-buffered saline, 0.05% Tween-20 (TBST) and probing with goat anti-mouse IRDye 800CW-conjugated antibody (Licor, Lincoln, NE, USA) for an additional 1 h. The membrane was then washed a further three times with TBST before imaging on a LI-COR Odyssey CLx imaging system.

### 2.13. Protection in K18-hACE2 Mice

Two mouse models were used to assess protection; (i) SARS-CoV-2_QIMR01_ (omicron BA.1) infection of homozygous K18-hACE2+/+ transgenic mice that were generated by inter-crossing K18-hACE2+/− mice and selecting for hACE2 homozygotes (manuscript in preparation), and (ii) SARS-CoV-2_QLD02_ (original strain) infection of heterozygous K18-hACE2+/− transgenic mice bred in-house by crossing with C57BL/6J mice (Animal Resources Center, Canning Vale, WA, Australia) and genotyping as described [26,28]. Mice were infected and tissue titers determined as described [19,28]. Briefly, mice received an intrapulmonary inoculum delivered via the intranasal route with stock virus from tissue culture supernatants diluted in RPMI 1640 with 5 × 10^4^ CCID_50_ of virus in 50 µL delivered to each lightly anesthetized mouse. At the indicated times, mice were euthanized and tissues were weighed, placed in medium, and homogenized twice at 6500 rpm for 15 s in a Precellys homogenizer. Tissues were kept on ice for 5 min before a 2nd homogenization, followed by centrifugation at 10,000 rpm (~9900× *g*) for 10 min. Virus titers in the supernatants were determined by standard CCID_50_ assays using Vero E6 cells [18] (using 96 well plates in quadruplicate) and expressed as log_10_CCID_50_/g. Mice were treated with the indicated purified mAb delivered by the i.p. route.

### 2.14. Statistics

Statistical analyses were undertaken using IBM SPSS Statistics for Windows, Version 19.0 (IBM Corp., Armonk, NY, USA). The t test was used if the difference in variances was <4, skewness was >−2 and kurtosis was <2. Otherwise the non-parametric Kolmogorov–Smirnov test was used.

## 3. Results

### 3.1. Generation of a Panel of Anti-SARS-CoV-2 Spike mAbs

Two C57BL/6J mice were vaccinated with sucrose cushion purified, UV-inactivated SARS-CoV-2_QLD02_, an isolate belonging to the original (ancestral) strain. A panel of mAbs were produced and characterized in a range of assay systems (Table 1). Although acetone-fixed SARS-CoV-2_QLD02_-infected Vero E6 cells were used for screening (with all the SARS-CoV-2 antigens presumably present, and confirmation of detection of nucleocapsid protein in addition to spike protein, Appendix A), all the virus-reactive mAbs generated herein recognized spike (Table 1).

### 3.2. Immunohistochemistry Using SCV2-1E8

Immunohistochemistry (IHC) was conducted on lung tissues from 3 mouse models and brain tissues from one mouse model, with mice infected with an omicron BA.1 isolate, SARS-CoV-2_QIMR01_, or an original strain isolate, SARS-CoV-2_QLD02_. A number of SARS-CoV-2 variants, including omicron, can use the mouse angiotensin-converting enzyme 2 (mACE2) receptor and are thus able to infect wild-type mice [34]. IHC using SCV2-1E8 of lungs from C57BL/6J mice infected with the omicron BA.1 isolate, clearly showing infection of bronchial epithelial cells (Figure 1a), consistent with previous reports [35,36].

The widely used K18-hACE2 transgenic mouse model expresses the SARS-CoV-2 receptor, human angiotensin-converting enzyme 2 (hACE2) from the keratin 18 promoter, and provides a severe model of COVID-19 [37,38]. Infection of these mice with SARS-CoV-2_QLD02_ resulted in a lethal outcome by 5 days post infection (dpi), with high viral titers found in lungs and brains [26,27,39]. IHC illustrated widespread staining of the alveolar epithelium (Figure 1b), consistent with previous reports [40,41]. IHC also showed staining across various regions of the brain (Figure 1c), with mortality in this model associated with brain infection [42]. The figure inset also shows the clear cytoplasmic staining of brain cells, with previous reports describing infection of neurons [42].

We have regenerated the transgenic mouse model described by Bao et al. [43], where hACE2 is expressed from the mouse ACE2 promoter, referred to herein as mACE-hACE2 mice [28]. IHC staining of bronchial epithelium is again readily apparent in this model (Figure 1d).

SCV2-1E8 thus emerges as a mAb showing excellent utility for IHC, clearly staining cells infected with two diverse variants of SARS-CoV-2 and in two different tissue types.

### 3.3. Indirect Immunofluorescent Assay Staining Using SCV2-6A11 and SCV2-1E8

Vero E6 cells have been widely used for in vitro infections with SARS-CoV-2 [18,44]. Indirect Immunofluorescent Assay (IFA) staining of SARS-CoV-2_QLD02_-infected Vero E6 cells using mAbs SCV2-6A11 and SCV2-1E8 and confocal microscopy shows clear patterns of cytoplasmic staining (Figure 2a,b). The IFA staining clearly distinguished between infected cells (Figure 2b, bottom left, green) and uninfected cells (Figure 2b, bottom left, U—blue DAPI staining with no green). Staining these infected cultures with a flavivirus-specific mAb as a negative control, instead of SCV2-6A11/SCV2-1E8, showed little or no signal (Figure 2c).

### 3.4. ELISAs

Standard ELISAs were undertaken using SARS-CoV-2 recombinant spike proteins from each variant of concern as antigens. These recombinant spike proteins were stabilized via six proline substitutions and mutated furin cleavage sites, and were expressed in Expi293-F™ cells [29,45]. The N-terminal domain (NTD) of the spike protein from the original strain was also used as an antigen [29]. ELISA analyses confirmed that the mAbs generated from these hybridoma fusions were directed to the spike protein. Three mAbs bound RBD, while eight had reactivity to NTD. Ten of the mAbs bound all variant spike proteins tested. A summary of the data generated from these experiments is shown in Table 1. Titration curves for purified mAbs used in other applications in this paper are shown in Figure 3a. For instance, the mAb used in Figure 1, SCV2-1E8, recognizes all variants of concern (but not the NTD) and has a IC50 of ≈0.1 µg/mL (Figure 3a). Titration curves for the remaining mAbs (Table 1) are shown in Appendix A. ELISA titrations using the recombinant Receptor Binding Domain (RBD) [29] as antigen are shown in Appendix A. The utility of several mAbs in fixed cell ELISAs are shown in Appendix A.

### 3.5. Western Blotting

The utility of several mAbs in Western blotting applications was assessed by detection of spike protein(s) in lysates of Vero E6 cells infected with the original strain isolate, SARS-CoV-2_QLD02_ (Table 1). Eight of the mAbs could detect spike in infected cells, with no detection of bands in uninfected cells (Table 1, Figure 3b, left). SCV2-6A11 and SCV2-1E12 also recognized unreduced spike (Figure 3b, right). These Western blots allow the target of these mAbs to be identified, either S1 or S2 (Figure 3, Reactivity), with SCV2-4C5 only able to detect reduced, full-length spike (Figure 3b, SCV2-4C5).

### 3.6. Lateral Flow Spike Antigen Detection Applications

SCV2-6A11 and SCV2-7E9 mAbs were assessed for their ability to capture and detect recombinant spike proteins, respectively in a lateral flow, rapid antigen test (RAT) system (Figure 4a). Spike antigen for the original, delta and omicron variants, provided visible positive bands at >100 ng (Figure 4b). At <50 ng the omicron Spike protein was no longer reliably detectable (Figure 4c).

### 3.7. Neutralization Assays

Neutralizing antibody titers were determined for a selection of purified mAbs using Vero E6 cells and a CPE-based assay in 96 well plates [19,21], with the flavivirus specific mAb 4G2 used as a negative control. Neutralization curves against an original strain isolate, SARS-CoV-2_QLD02_, are provided; SCV2-5A1 and SCV2-3H9 showed the highest potency with the concentrations of mAb providing 50% neutralization (IC50) at ≈3 and 0.35 µg/mL, respectively (Figure 5). SCV2-7E1/4 also showed neutralizing activity; however, yield from this hybridoma was low (Appendix A). Of those mAbs that were neutralizing, mAb SCV2-7E9 showed the lowest potency with an IC50 of 14.2 µg/mL (Figure 5).

### 3.8. Protection against Omicron and Original Strain Isolates in K18-hACE2 Mice

MAbs are being widely considered for the treatment of COVID-19 both for pre-exposure prophylaxis and post-exposure therapy, with antibodies that can target multiple variants of concern clearly favored [46,47]. Purified mAbs with the best neutralizing activities, SCV2-5A1 and SCV2-3H9 (Figure 5), were used to treat K18-hACE2 mice infected with SARS-CoV-2 viruses (Figure 6). From the ELISA data Kd estimates were 1.04 nM for SCV2-5A1 (BA.1 omicron), and 1.35 nM for SCV2-5A1 and 1.35 nM for SCV2-3H9 (original).

Treatment of mice infected with a BA.1 omicron isolate, SARS-CoV-2_QIMR01_ [15], with a single 1 mg dose of SCV2-5A1 (Figure 6a) resulted in a significant reduction in lung titers of ≈1.5 logs (Figure 6b). Reduction of viral titers in the nasal turbinates did not reach significance (Figure 6b). The 2 dpi time point was chosen as this represents the peak in lung virus titers in this non-lethal model.

Treatment of mice infected with an original strain isolate, SARS-CoV2_QLD02_ with SCV2-5A1 and SCV2-3H9, was also investigated, with the 5 dpi time point chosen as this represents the ethically defined end point in this lethal model (Figure 6c). Therapeutic treatment significantly reduced the infection-associated weight loss (Figure 6d) and reduced nasal turbinate and brain titers to below detection, but had no significant effect on lung titers (Figure 6e). Prophylaxis also significantly reduced weight loss (Figure 6f), and significantly reduced viral titers in all tissues tested (Figure 6g). (We also illustrate in this experiment that PBS behaved similarly to the 4G2 control mAb; Figure 6f,g). These results are broadly comparable with previous studies evaluating anti-spike mAbs in COVID-19 mouse models [48,49,50,51].

## 4. Discussion

Since the outbreak of SARS-CoV-2 and the ensuing pandemic, a sizable number of mAbs have been generated for various applications [5,48,50,51,52,53,54]. Perhaps the standout mAb described herein is the RBD-binding, SCV2-1E8, which showed excellent performance in IHC and recognized multiple variants of concern. In addition, SCV2-1E8 also worked well in IFA, ELISA and Western blotting applications.

Rapid antigen tests (RAT) have been developed by a number of companies [55,56]. Detection of nucleocapsid is often used in these tests due to lower levels of sequence variation [57]. Nevertheless, the spike-specific mAb pair described herein, SCV2-6A11 and SCV2-7E9, could detect multiple variants of concern, suggesting they recognize a conserved region of the spike protein. The sensitivity of RATs are usually determined relative to standard RT-qPCR tests [57], rather than the ng of spike protein used herein. The ability to detect 100 ng of Spike is arguably at the lower end of sensitivity of commercial RATs, with multiple avenues for optimization generally pursued to improve sensitivity (e.g., antibody coating, gold conjugation, buffer composition).

Several clinical trials have established the safety and some efficacy of mAb treatments [14,58,59], although optimal therapeutic efficacy generally requires early administration to COVID-19 patients [60]. Given the rapid and robust nature of the K18-hACE2 mouse model, antiviral activity of mAb treatment is often demonstrated in this model by administering the mAb prior to infection [48,49,50,52], although some mAb cocktails have shown significant anti-viral activity if given 24 h after infection [50]. Antibody escape presents a salient problem as the virus continues to evolve [14,61,62], so mAbs that retain activity across multiple variants of concern are clearly of higher utility [63]. For instance, tixagenvimab and cilgavimab (Evusheld, AZD7442) retained their neutralizing activity against omicron BA.1 and showed protective activity in K18-hACE2 mouse studies, with results comparable to those presented herein [64]. SCV2-5A1 showed protection against both an original strain and an omicron variant (Figure 6b). Development of a commercial product would now require affinity-maturation [65,66] and bioengineering inter alia to humanize the antibody, to reduce the potential risk of antibody-dependent enhancement and to optimize pharmacokinetics [67].

In summary, we have generated mAbs, some of which have potential for further development in diagnostic and treatment scenarios. Perhaps of most immediate value is SCV2-1E8, a mAb suitable for multiple research applications, including excellent performance in IHC, and capable of recognizing multiple variants of concern.

## Figures and Tables

**Figure 1 viruses-15-00139-f001:**
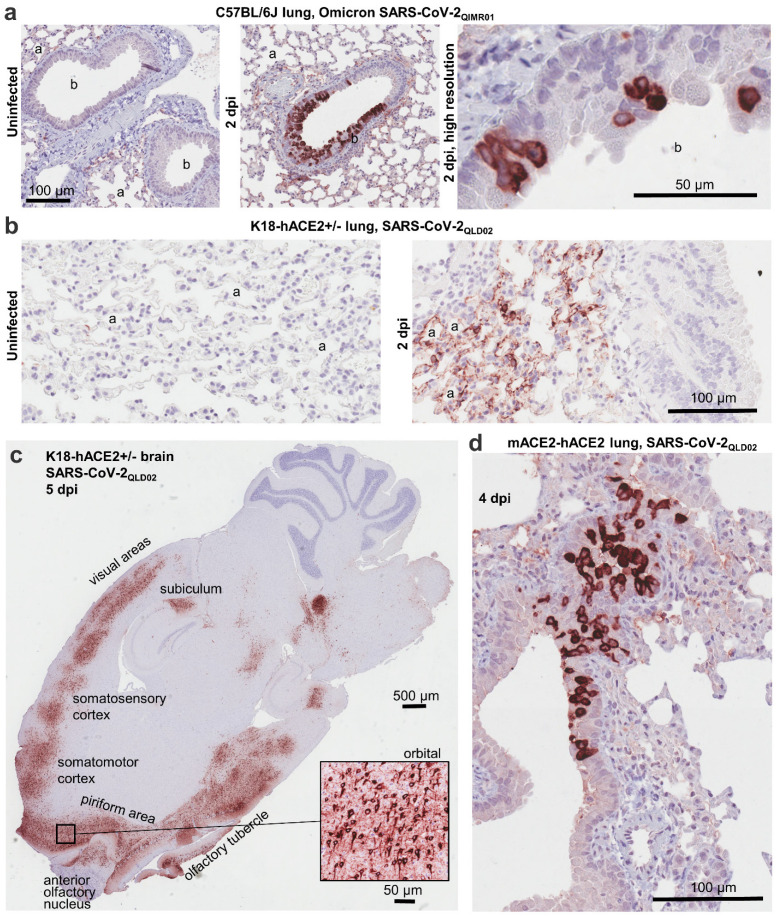
Immunohistochemistry using SCV2-1E8. (**a**) Uninfected lungs (left) and infected lungs at 2 dpi (two images on the right) from C57BL/6J mice infected with an omicron isolate, SARS-CoV-2_QIMR01_. (**b**) Uninfected lungs (left) and infected lungs at 2 dpi (right) from K18-hACE2^+/−^ mice infected with an original strain isolate, SARS-CoV-2_QLD02_. (**c**) Brain from K18-hACE2^+/−^ mice on 5 dpi with SARS-CoV-2_QLD02_. Stained areas of the brain are annotated (approximate); insert taken from the orbital region of the brain. (**d**) Infected lungs 4 dpi from mACE2-hACE2 mice infected with SARS-CoV-2_QLD02_. Within the IHC panels of (a) and (b), alveolar spaces (a) and bronchioles (b) are indicated.

**Figure 2 viruses-15-00139-f002:**
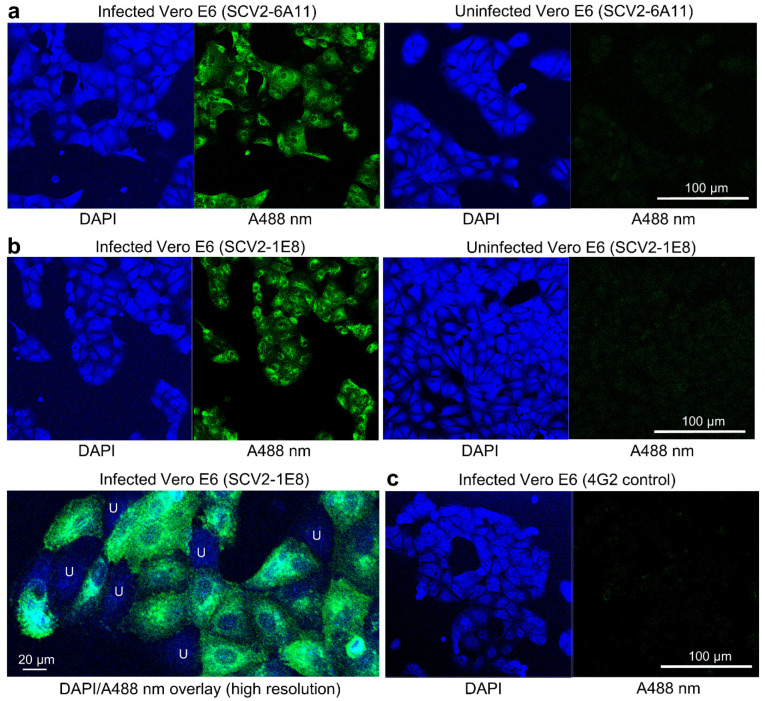
Indirect Immunofluorescence Assay (IFA) staining. (**a**) Vero E6 cells infected with SARS-CoV-2_QLD02_ (left) or uninfected (right) stained with DAPI (blue) and anti-SARS-CoV-2 mAb, SCV2-6A11 (green, A488 nm). (**b**) As in a, but stained with SCV2-1E8. Bottom left shows a high resolution image with DAPI/A488 nm overlap, where infected cells (green) can clearly be distinguished from uninfected cells (blue, labelled U). (**c**) Vero E6 cells infected with SARS-CoV-2_QLD02_ stained with a negative control mAb, 4G2.

**Figure 3 viruses-15-00139-f003:**
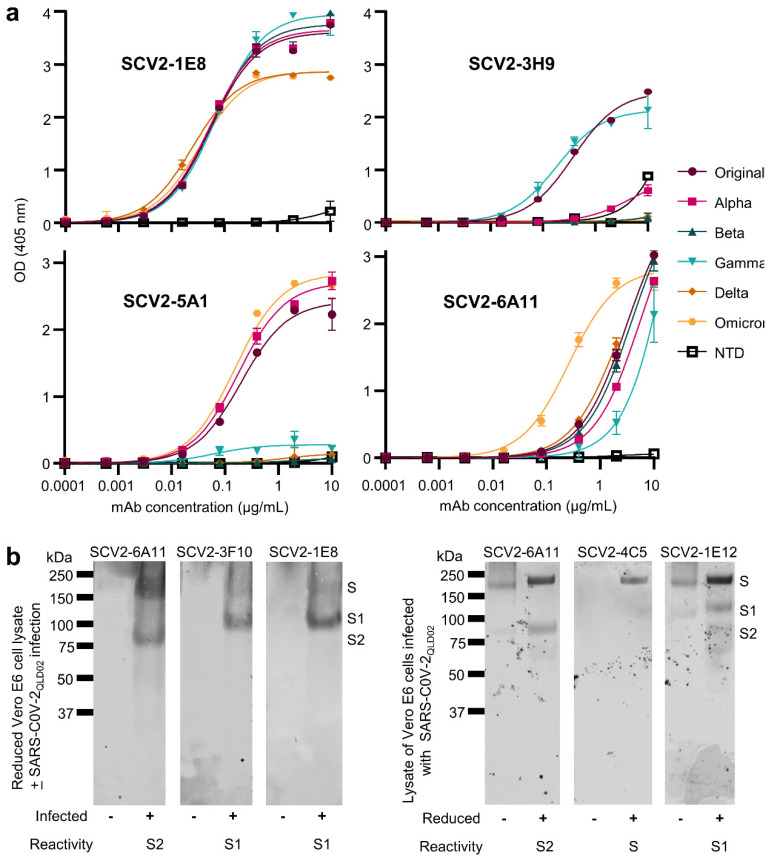
ELISAs and Western blots. (**a**) ELISA curves for the indicated purified mAbs using recombinant HexaPro spike proteins with mutated furin cleavage site from the indicated variants of concern as antigen. (**b**) Western blots using reduced (DDT) lysates of Vero E6 cells with (+) and without (−) infection with SARS-CoV-2_QLD02_ (left), or lysates of SARS-CoV-2_QLD02_ infected Vero E6 cells that were either reduced (+) or not reduced (−) (right). The primary reactivity of each mAb is indicated. (Note SCV2-6A11 is used in both settings).

**Figure 4 viruses-15-00139-f004:**
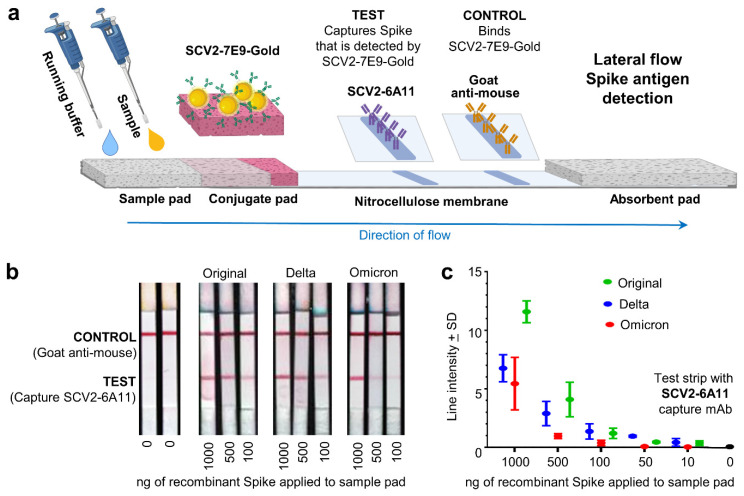
Lateral flow spike antigen detection applications. (**a**) Setup of the lateral flow spike antigen detection system (generated using BioRender). Sample (containing recombinant spike) is applied followed by running buffer, which carries the antigen and mobilizes the gold-conjugated SCV2-7E9 mAb. Spike bound to SCV-6A11 (the capture mAb) is then detected by SCV2-7E9-Gold. The control binds the SCV2-7E9-Gold illustrating that the detection-mAb has been mobilized. (**b**) The SCV2-7E9-Gold is visualized as a red band that illustrates the presence of recombinant spike in the sample (TEST), with a red CONTROL line illustrating that the detection mAb (SCV2-7E9-Gold) has been mobilized. (**c**) Quantitation of antigen detection by the lateral flow test for recombinant spike antigens from each of 3 variants of concern. (SD from n = 3 replicates).

**Figure 5 viruses-15-00139-f005:**
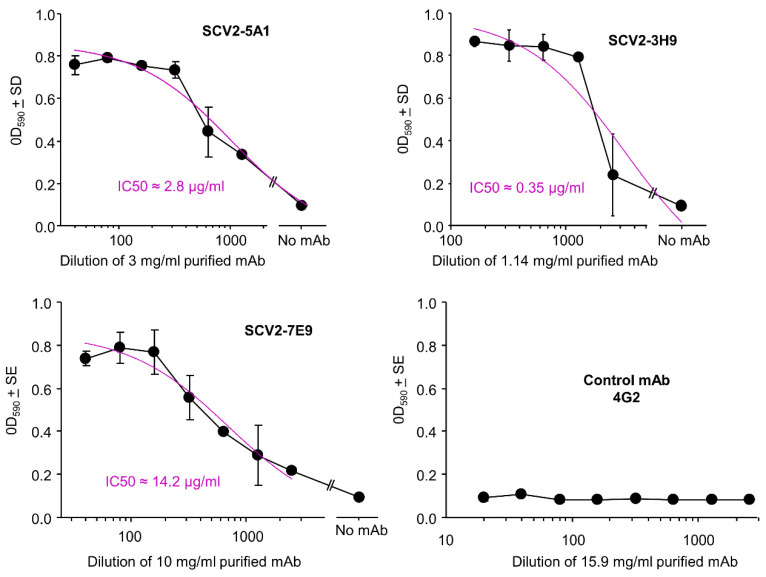
Virus neutralization assays. The indicated concentrations of purified antibodies were serially diluted in duplicate and used to neutralize SARS-CoV-2_QLD02_ in Vero E6 CPE-based assays. High OD values illustrate cells protected from CPE by mAbs and thus staining with crystal violet, and low OD values represents a loss of cells via CPE resulting in loss of crystal violet staining; 50% inhibitory concentrations (IC50) were determined by interpolation. 100% CPE (0% neutralization) was determined in 8 replicates in wells without mAb. IC50 values were determined using the “log(inhibitor) vs. response” feature of GraphPad Prism (purple).

**Figure 6 viruses-15-00139-f006:**
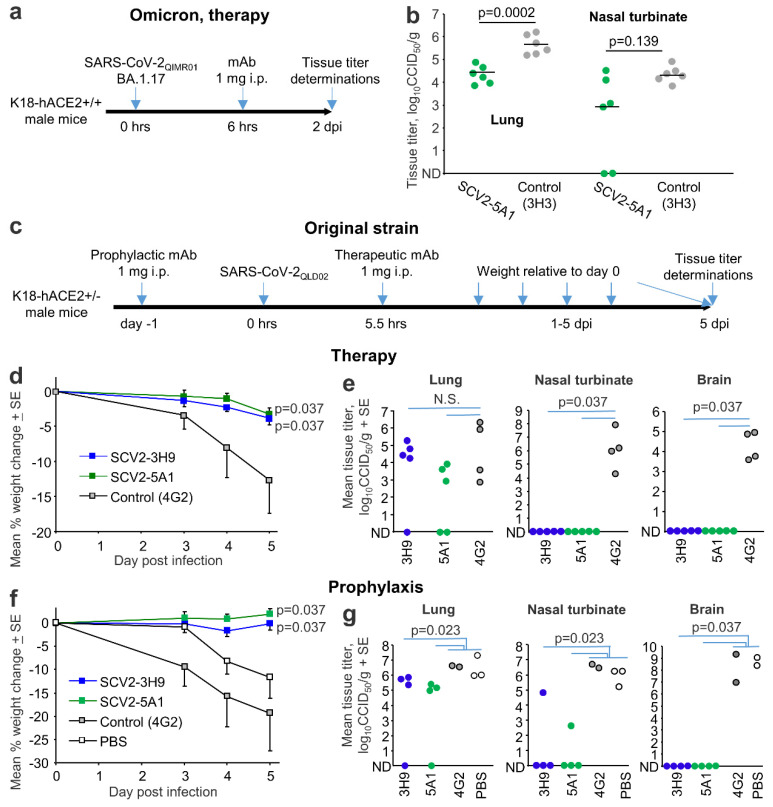
Protection in K18-ACE2 mice. (**a**) Timeline of mAb therapy of K18-ACE2^+/+^ mice infected with a BA.1 omicron isolate (SARS-CoV-2_QIMR01_). The anti-SARS-CoV-2 antibody was SCV2-5A1 and the control mAb was an isotype control, anti-Zika virus NS1 (3H3). (**b**) Lung and nasal turbinate titers for mice described in a. Statistics by t test (lung) and Kolmogorov–Smirnov test (nasal turbinates). (**c**) Timelines for prophylactic and therapeutic treatment of K18-ACE2^+/−^ mice infected with an original strain isolate (SARS-CoV-2_QLD02_) with SCV2-5A1 or SCV2-3H9, or the control mAb, anti-flavivirus E (4G2). (**d**) Mean weight change relative to 0 dpi after mAb therapy. Statistics by Kolmogorov–Smirnov tests for 5 dpi. (**e**) Tissue titrations for the same mice shown in d (5 dpi). Statistics by Kolmogorov–Smirnov tests. N.S.—not significant. (**f**) Mean weight change relative to 0 dpi after mAb prophylaxis. For statistics, data for 4G2 and PBS were combined to represent the controls (n = 5). Statistics by Kolmogorov–Smirnov tests on 5 dpi. (**g**) Tissue titrations for the same mice are shown in f (5 dpi). Statistics by Kolmogorov–Smirnov tests, with 4G2 and PBS groups combined.

**Table 1 viruses-15-00139-t001:** List of anti-SARS-CoV-2 spike mAbs, characterization and behavior in assays systems.

mAb SCV2-	Isotype	Western Blot Original ^a^	ELISA ^b^ Original	ELISA Alpha	ELISA Beta	ELISA Gamma	ELISA Delta	ELISA Omicron	rRBD ELISA Original ^c^	rNTD ELISA Original ^d^	Neutralization Original ≈IC50 µg/mL
1E8	IgG2a	+ (S1)	+	+	+	+	+	+	+/−	−	-
3F10	IgG2a	+ (S1)	+	+	+	+	+	+	−	+	-
6A11	IgG2a	+ (S2)	+	+	+	+	+	+	−	−	-
1E12	IgG2b	+ (S1)	+	+	+	+	+	+	−	+	-
1F1	IgG2b	−	+	+	+	+	+	+	−	+	-
1F9	NT	−	+	+	+	−	+	+	−	+/−	-
2B8	NT	+ (S1)	+	+	+	+	+	+	−	−	-
2F3	NT	−	+	+	+	−	+	+	+	−	NT
3C7	NT	+ (S1)	+	+	+	+	+	+	−	+	NT
3H9	IgG2b	−	+	+/−	−	+	−	−	−	+/−	0.35
4C5	NT	+ (S)	+	+	+	+	+	+	−	+	-
5A1	IgG2b	−	+	+	−	−	−	+	−	−	2.8
7C5	NT	+ (S1)	+	+	+	+	+	+	−	+	NT
7E1/4	IgG2b	−	+	+	+	+	+	+	+	−	≈8 ^e^
7E9	IgG2a	−	+	+	−	+	−	+	−	−	14.2

^a^ Using lysates of Vero E6 infected with original strain isolate, SARS-CoV-2_QLD02_. ^b^ ELISAs were undertaken using recombinant HexaPro spike proteins with mutated fusion cleavage sites, and containing the characteristic changes associated with the indicated variants of concern. ^c^ Tested at mAb concentration of 100 µg/mL (RBD from original Wu-Hu-1 isolate). ^d^ Tested at mAb concentration of 10 µg/mL; +/− equivocal result (spike N-terminal domain (NTD) from original strain, Wu-Hu-1 isolate). ^e^ Estimate only; low mAb yield from hybridoma, tested using single wells (i.e., not in duplicate). NT—not tested.

## Data Availability

The data presented in this study are available within the article and Appendix A.

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
