# Peer review of "Monoclonal Antibodies Specific for SARS-CoV-2 Spike Protein Suitable for Multiple Applications for Current Variants of Concern"

_viruses, 2022, doi:10.3390/v15010139_

Round 1

Reviewer 1 Report

This paper describes the identification and characterization of a panel of new mouse monoclonal antibodies against SARS-CoV-2 Spike protein with the use of multiple imaging, immunological and biochemical technics. The experiments are properly controlled and the results look very clear. Here are just a few minor corrections/comments:

1) The powers of 10 should be written in superscript

2) Line 274 “the t-test” (by the way, I’m not sure that considering the small number of individuals in figure 6b, the t-test is appropriate)

3) Line 355 “The utility of several mAbs in fixed cells ELISAs are shown…”

4) Line 369: please remove “were”

5) Since the mice have been immunized with inactivated virus, one could have expected that antibodies against other viral proteins (M, N) would also be generated. Were there none or did the author only concentrated on anti-S?

Author Response

Reviewer 1

 1) The powers of 10 should be written in superscript

     Corrected throughout

2) Line 274 “the t-test” (by the way, I’m not sure that considering the small number of individuals in figure 6b, the t-test is appropriate)

     Arguably the limit for t test is n=3, but here we have n=5 per group, so a t test would normally be viewed as suitable.  Furthermore, all controls are higher than all mAb treated, so a Mann Whitney U test (equal variance) would give a very similar answer, p=0.008

3) Line 355 “The utility of several mAbs in fixed cells ELISAs are shown…”

     Corrected

4) Line 369: please remove “were”

     Corrected

5) Since the mice have been immunized with inactivated virus, one could have expected that antibodies against other viral proteins (M, N) would also be generated. Were there none or did the author only concentrated on anti-S?

     We have added the sentence “Although acetone-fixed SARS-CoV-2QLD02-infected Vero E6 cells were used for screening, all virus reactive mAbs recognized spike (Table 1)” to 3.1 Results to explain our “focus” on spike.

Reviewer 2 Report

Point 1: Figure 2, all the pictures have no scales.

Point 2: Line 315, if you want to elucidate that SCV2- 1E8 can be applied in IHC well, you would better tell us the working concentration in IHC.

 Point 3: 105 per 24 well in line152, Vero E6 at 104/well in line171 and 5x104 CCID50 of virus in line263, these numbers labeled in yellow should be superscript.

 Point 4: Line 352, IC50 (50% inhibitory concentrations) is used to describe the dose of the drugs inhibit the 50% of activity of the causative agent.Herein IC50 is not correct.

 Point 5: Figure 5, the IC50 of SCV2-5A1 and SCV2-3H9 should be calculated in software GraphPad Prism through the model inhibitor vs. response in nonlinear regression (curve fit).

 Point 6: Figure 6a&6c, in the post-exposure therapy, the therapeutic mAbs were administrated in 6hrs after SARS-CoV-2QIMR01 infection and 5.5hrs after SARS-CoV-2QLD02 infection, respectively, why chose these timepoints? To my knowledge, most SARS-CoV-2 neutralizing antibodies was administrated in 24hrs after SARS-CoV-2 infection, have you tried this timepoint?

 Point 7: Figure 6, in the pre-exposure prophylaxis assay, PBS group was set up, but in the post-exposure therapy assay, the PBS group was abolished, please explain the reason.

 Point 8: Figure 6g, for the tissue titer in the lung and nasal turbinate, there are 3 mice in the PBS group, but there are 2 mice in the PBS group in Brain tissue titer, please explain the reason.

 Point 9: In Fig6 a, the authors choose 2 dpi in omicron strain to determine tissue titers. In Fig 6c for original strain therapy, the authors choose 5 dpi to determine tissue titers. Please provide your reasonable explanation.

 Point 10: The authors developed an IHC using SCV2-1E8, but in the part of Protection in K18-hACE2 mice, the authors haven’t provide the IHC results using the developed method. Generally the protective antibody should provide protection in the lungs and brains.

 Point 11As we know, a protective neutralizing antibody must have good KD, the authors haven’t provide the mAb 5A1 and mAb 3H9 specific binding ability to the antigen.

Author Response

Reviewer 2

Point 1: Figure 2, all the pictures have no scales.

     Scale bars have been added

 Point 2: Line 315, if you want to elucidate that SCV2- 1E8 can be applied in IHC well, you would better tell us the working concentration in IHC.

     The concentration of IgG in the hybridoma supernatant is provided in the Methods

  Point 3: 105 per 24 well in line152, Vero E6 at 104/well in line171 and 5x104 CCID50 of virus in line263, these numbers labeled in yellow should be superscript.

     Corrected throughout

 Point 4: Line 352IC50 (50% inhibitory concentrations) is used to describe the dose of the drugs inhibit the 50% of activity of the causative agent.  Herein IC50 is not correct.

     IC50 is also widely used for antibodies (see refs below); IC50 in this context is the concentration of mAb that provides 50% neutralization (inhibition of virus replication).

Nature volume 593, pages130–135 (2021)

Nature volume 602, pages671–675 (2022)

Nature Immunology volume 22, p 1503–1514 (2021)

Cell Rep. 2020 Jul 21;32(3):107918

We have clarified the definition of IC50 in the text as it pertains to antibodies. 

 Point 5: Figure 5, the IC50 of SCV2-5A1 and SCV2-3H9 should be calculated in software GraphPad Prism through the model inhibitor vs. response in nonlinear regression (curve fit).

     Curve fits have been added to the figures as requested and the IC50s from Prism provided.

 Point 6: Figure 6a&6c, in the post-exposure therapy, the therapeutic mAbs were administrated in 6hrs after SARS-CoV-2QIMR01 infection and 5.5hrs after SARS-CoV-2QLD02 infection, respectively, why chose these timepoints? To my knowledge, most SARS-CoV-2 neutralizing antibodies was administrated in 24hrs after SARS-CoV-2 infection, have you tried this timepoint?

     Have rephrased this line to make this issue clearer “Given the rapid and robust nature of the K18-hACE2 mouse model, antiviral activity of mAb treatment is often demonstrated in this model by administering the mAb prior to infection [46-48, 50], although mAb cocktails have shown significant anti-viral activity if given 24 hours after infection [50]”. 

 Point 7: Figure 6, in the pre-exposure prophylaxis assay, PBS group was set up, but in the post-exposure therapy assay, the PBS group was abolished, please explain the reason.

     The control mAb is the critical control, and 6f shows it is not significantly different from PBS.  We have added the sentence to the results; “We also illustrate in this experiment that PBS behaved similarly to the 4G2 control mAb; Fig. 6f,g”. 

 Point 8: Figure 6g, for the tissue titer in the lung and nasal turbinate, there are 3 mice in the PBS group, but there are 2 mice in the PBS group in Brain tissue titer, please explain the reason.

     We were unable to purify sufficient 4G2 to the same standard as the other mAbs.  

 Point 9: In Fig6 a, the authors choose 2 dpi in omicron strain to determine tissue titers. In Fig 6c for original strain therapy, the authors choose 5 dpi to determine tissue titers. Please provide your reasonable explanation.

     We have added two sentences to explain these time points “The 2 dpi time point was chosen as this represents the peak in lung virus titers in this non-lethal model.”  “the 5 dpi time point chosen as this represents the ethically defined end point in this lethal model”.

 Point 10: The authors developed an IHC using SCV2-1E8, but in the part of Protection in K18-hACE2 mice, the authors haven’t provide the IHC results using the developed method. Generally the protective antibody should provide protection in the lungs and brains.

     Viral titrations are much more sensitive than IHC as a readout of brain infection, with IHC not suitable as a quantitative measure of protection.  We agree that protection in lungs and brain would be ideal; however, our experience suggest it is easier to prevent brain infection than lung infection (see ref 37).  Presumably this is because brain requires transit across the olfactory epithelium, whereas challenge virus is inoculated directly into the lungs; however, this constitutes some considerable speculation which is probably not warranted.

 Point 11As we know, a protective neutralizing antibody must have good KD, the authors haven’t provide the mAb 5A1 and mAb 3H9 specific binding ability to the antigen.

     Kd estimates have been added to the Results.